

# Mangrove ecosystem properties regulate high water levels in a river delta

Ignace Pelckmans[1], Jean-Philippe Belliard[1,2], Luis E. Dominguez-Granda[3], Cornelis Slobbe[4], Stijn Temmerman[1], Olivier Gourgue[1,5,2]

[1]ECOSPHERE, University of Antwerp, Department of Biology, Antwerp, Belgium
[2]Royal Belgian Institute of Natural Sciences, Brussels, Belgium
[3]Centro del Agua y Desarrollo Sostenible, Escuela Superior Politecnica del Litoral (ESPOL)*, Faculdad de Ciencias Naturales y Matematicas, Guayaquil, Ecuador
[4]Geoscience & Remote Sensing, Delft University of Technology (TU Delft), Delft, Netherlands
[5]Department of Earth and Environment, Boston University, Boston, MA, USA

*Correspondence to*: Ignace Pelckmans (ignace.pelckmans@uantwerpen.be)

**Abstract.** Intertidal wetlands, such as mangroves in the tropics, are increasingly recognized for their role in nature-based mitigation of coastal flood risks. Yet it is still poorly understood how effective they are in attenuating the propagation of extreme sea levels through large (order of 100 km$^2$) estuarine or deltaic systems, with complex geometry formed by networks of branching channels intertwined with mangrove and intertidal flat areas. Here, we present a delta-scale hydrodynamic modelling study, aiming to explicitly account for these complex landforms, for the case of the Guayas delta (Ecuador), the
largest estuarine system at the Pacific coast of Latin America. Despite coping with data scarcity, our model accurately reproduces the observed propagation of high water levels during a spring tide. Further, based on a model sensitivity analysis, we show that high water levels are most sensitive to the mangrove platform elevation and degree of channelization, but to a much lesser extent to vegetation-induced friction. Mangroves with a lower surface elevation, lower vegetation density and higher degree of channelization all favour a more efficient flooding of the mangroves and therefore more effectively attenuate
the high water levels in the deltaic channels. Our findings indicate that vast areas of channelized mangrove forests, rather than densely vegetated forests, are most effective for nature-based flood risk mitigation in a river delta.

## 1 Introduction

Low-lying coastal areas, such as river deltas, are hotspots of human activity, but their low elevation makes them particularly vulnerable to coastal flood hazards from extreme sea level events, driven by events such as storm surges and climate
fluctuations. With global warming, these events are expected to increase in intensity and frequency and together with long-term sea level rise and land subsidence, coastal flood risks are expected to increase in the future (Day et al., 2016; Fox-Kemper et al., 2021). Material and human damages are therefore expected to increase dramatically with projected economic losses up to nearly 10 % of the global GDP by 2100 (Neumann et al., 2015; Tessler et al., 2015). As a consequence, there is a growing need for innovative science-based strategies to mitigate flood risks in low-lying coastal areas (Glavovic et al., 2022).



Nature-based flood risk mitigation is one approach that has gained particular interest over the past decade (Narayan et al., 2016; Temmerman et al., 2013). Within river deltas and estuaries, tidal wetlands such as mangroves and tidal marshes, can potentially attenuate extreme sea levels when propagating upstream (Guannel et al., 2016; Wamsley et al., 2010). Extreme sea levels, such as storm surges or anomalously high tides, propagate through deltas and estuaries as long waves, of which the top of the wave (i.e. the high water level) can be reduced by the presence of tidal wetlands, through two mechanisms (Temmerman

et al., 2022). Firstly, as high water levels propagate through continuous unchanneled wetlands, vegetation-induced drag limits the transport of water, hence causing high water levels to lower with distance travelled through the wetland. This first mechanism is further referred to as '*within wetland attenuation*' (Krauss et al., 2009; Stark et al., 2015). Secondly, as water levels rise above the channel banks, water flows laterally into the wetlands where it is spread out and temporarily stored, as such lowering upstream high water levels: This second mechanism is further referred to as '*along-estuary* attenuation'

(Smolders et al., 2015). For mangroves, few small-scale empirical observations during extreme sea level events have quantified attenuation rates, which are typically expressed as high water level reduction per distance travelled by the flood wave. Reported values range between 0 and 36 cm/km depending on the type of the high water event (e.g., spring tide, storm surge) and of the wetland ecosystem (Horstman et al., 2021; J. Montgomery et al., 2018; Stark et al., 2015).

Variations in the rate of high water level attenuation can be partly related to variations in the wetland vegetation properties.

Vegetation induces drag on water flow, which has been shown empirically to depend on vegetation properties such as stem width and stem density (Mazda et al., 1997, 2005; Vandenbruwaene et al., 2013). More recently, numerical models based on the shallow water equations provided insights on the effect of vegetation properties regarding the propagation of extreme high water levels (Chen et al., 2021; Stark et al., 2016; Zhang et al., 2012). The vegetation-induced drag is typically included as an additional sink term in the flow momentum equations (Baptist et al., 2007). For mangroves, the latter is typically parameterized

as a function of a drag coefficient, representing the roughness of a single mangrove stem or prop root, and the density of stems and roots, quantified as their frontal surface area (Horstman et al., 2015). Model simulations have demonstrated that within-wetland attenuation of high water levels is stronger for higher simulated vegetation-induced drag (Chen et al., 2021; Mori et al., 2022).

Furthermore, in addition to vegetation properties, the wide range of observed and modelled attenuation rates can be partly

explained by variations in the wetland platform topography, more specifically by the wetland platform elevation and degree of channelization. Firstly, within-wetland attenuation has been shown to decrease with inundation depth, based on observations both in marshes (Glass et al., 2018; Stark et al., 2015) and mangroves (Horstman et al., 2021). Hydrodynamic models have confirmed and explained this due to a reduced effect of the bottom friction (Montgomery et al., 2019). However, for along-estuary attenuation in marsh-dominated systems, model simulations showed the opposite: the lower the wetland platform the

higher the along-estuary attenuation rates as a larger fraction of the incoming flood water volume can be laterally spread out and temporarily stored in the wetlands fringing the estuarine channel (Smolders et al., 2015). Secondly, tidal channels, which typically dissect wetlands, allow for a more rapid flood propagation (Horstman et al., 2015, 2021; Vandenbruwaene et al., 2015). In channelized mangroves, attenuation rates are negligible (Montgomery et al., 2018). For marshes, observations



showed the highest attenuation rates in non-channelized continuous portions of the marsh, while attenuation rates were lower when measured along channels, and they decreased with increasing channel width (Stark et al., 2015).

Current understanding of the role of wetland vegetation properties and wetland topography on extreme water level attenuation is based on either empirical observations on relatively small scales ($\sim$10²-10³ m) (Horstman et al., 2021; Krauss et al., 2009; J. Montgomery et al., 2018), or on hydrodynamic models that may include larger scales but often with relatively simplified geometry e.g. (Chen et al., 2021; Zhang et al., 2012). The latter typically consider flood propagation through a continuous belt of mangroves or along a single estuarine channel fringed by mangroves (Chen et al., 2021; Deb & Ferreira, 2017; Dominicis et al., 2023; Smolders et al., 2015; Willemsen et al., 2016). To our knowledge, no studies exist that consider large scale (order of 100 km$^2$) river deltas, accounting for the effects of the complex geometry formed by networks of branching channels, varying in size from wide (order of 10³ m) to small (order of 10 m), and intertwined with vegetated and unvegetated intertidal areas. Hence, despite the fact that river deltas are hotspots of particularly high vulnerability to extreme sea level events, there is poor understanding on how high water level propagation is affected by the intrinsic complex bio-geomorphic nature of large river deltas. This is particularly true for tropical river deltas in low-income countries, where data are scarce but where there is a high potential for wetlands to act as nature-based strategies for coastal hazard mitigation (Temmerman et al., 2013).

Here, we aim to contribute to fill this knowledge gap, by calibrating and validating a hydrodynamic model of the Guayas delta, Ecuador, explicitly including the intertidal mangrove forests and the bare intertidal mudflats, as well as the complex channel networks that dissect them. This model is subsequently used to numerical model simulations allowing to unravel the relative importance of wetland vegetation properties, wetland platform elevation, topography of bare mudflats and degree of channelization inside the mangrove forests, in affecting the spatial distribution of high water levels at the scale of the entire delta.

## 2. Methods

A brief overview of the methodological approach is presented below. Further details are described in the Supporting Information, as indicated in several places below.

### 2.1 Study Area

The Guayas delta (Fig. 1A) is the largest river delta along the Pacific coast of Latin America (Twilley et al., 2001). The delta consists of 2 major branches (Fig. 1B). The eastern branch, the Guayas channel, receives freshwater discharge from the Guayas river which is formed at the confluence of the Babahoyo and Daule river. Its discharge is characterised by a strong seasonality ranging from about 200 m$^3$/s in the dry season, from April to November, up to about 1600 m$^3$/s in the wet season, from December to March (INHAMI, 2019). The western branch, Estero El Salado, does not have any significant freshwater input.



At the seaside, the delta is connected to the Gulf of Guayaquil from where semidiurnal tides enter the delta with a tidal range of about 2 m (Fig. 1A). When propagating upstream through the delta, the tidal range increases up to about 5 m near the city of Guayaquil.

Mangroves naturally cover the deltaic plain. They are mostly dominated by one species, *Rhizophora mangle*, with locally young mangrove patches of *Avicennia germinans* (Hamilton, 2019). However, since the 1960s, large areas of mangroves have been converted into aquaculture ponds, essentially for shrimp farming (Hamilton, 2019). In the northern part of the delta lies the city of Guayaquil (Fig. 1A), Ecuador's largest and economically most important city. According to a global assessment, the city ranks fourth among most vulnerable cities to coastal flood hazards (Hallegatte et al., 2013). El Niño events are the main source of extreme high water levels and flood risks. For instance, during the particularly strong El Niño event of 1997-1998 that lasted over 18 months, high water levels were on average 40 cm higher than during neutral climate conditions in the inner delta, and even reached up to 1 m higher when the El Nino event was most intense (Belliard et al., 2021). Vice versa, high water levels in the delta can decrease during strong La Niña events for several decimeters (Belliard et al., 2021). Its low-lying position and the high concentration of socio-economic activities make the Guayas delta a typical example of a tropical delta where impacts of sea level rise and intensification of climate fluctuations as El Niño Southern Oscillation (ENSO) will drastically increase in the coming decades.



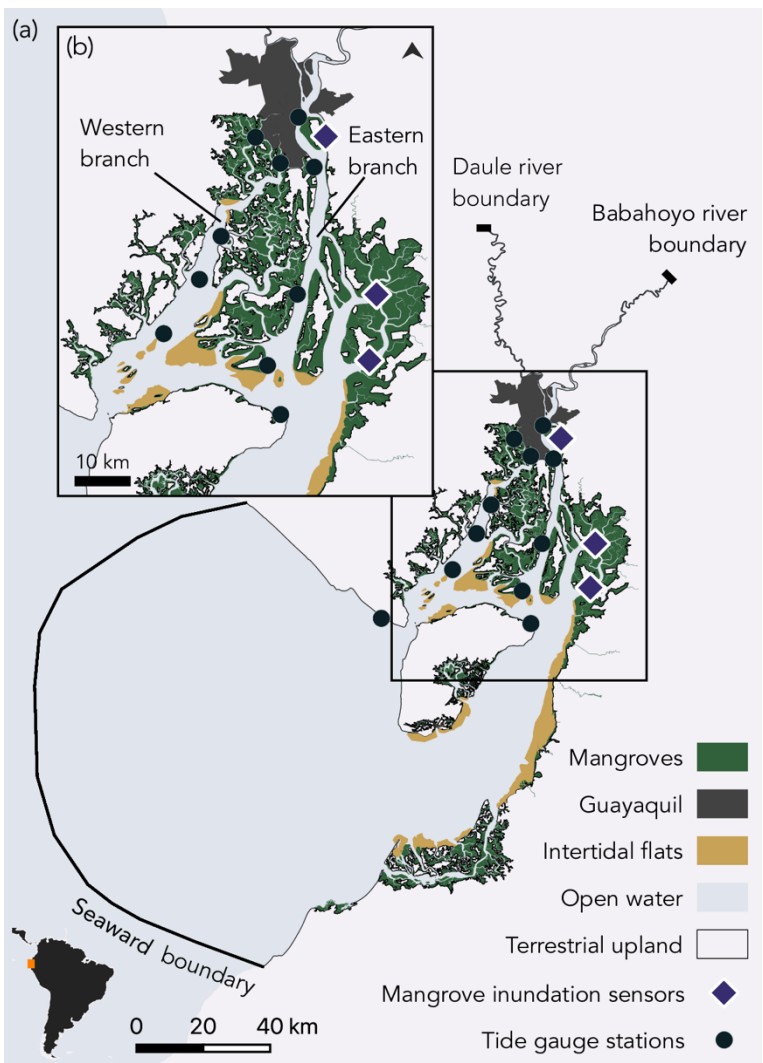

**Figure 1. Map showing the Gulf of Guayaquil (a) and Guayas delta (b), and indicating the area included in the model domain and positions of the seaward and upstream model boundaries. Large intertidal zones, mangroves (green) and intertidal flats (yellow), are spread over the delta and border the subtidal areas (light blue) and together form the model domain. Orange markers indicate tide gauge stations and red markers indicate mangrove inundation sensors.**

## 2.2 Model equations

To model the hydrodynamics, we used TELEMAC 2D (v8p2r0), which is part of the open-source finite element solver suite of Telemac (Hervouet, 2007). The governing equations are the depth-averaged shallow water equations:

$$\frac{\partial h}{\partial t} + \boldsymbol{\nabla} \cdot h\boldsymbol{v} = 0 \qquad (1)$$

$$\frac{\partial \boldsymbol{v}}{\partial t} + \boldsymbol{v} \cdot \boldsymbol{\nabla} \boldsymbol{v} = -g\boldsymbol{\nabla}\eta + \frac{1}{h}\boldsymbol{\nabla} \cdot (h\nu\boldsymbol{\nabla}\boldsymbol{v}) - \frac{\boldsymbol{\tau_b} + \boldsymbol{\tau_v}}{\rho h} \qquad (2)$$



where $h$ is the water depth (m), $\nabla$ is the differential operator (/m), $t$ is the time (s), $\boldsymbol{v}$ is the depth-averaged flow velocity (m/s), $g$ equals 9.81 m/s$^2$ is the gravitational acceleration, $\eta$ is the water surface elevation above a reference level (m), $\nu$ equals 0.01 m$^2$/s and is the diffusion coefficient, $\boldsymbol{\tau_b}$ is the bed shear stress (N/m$^2$), $\boldsymbol{\tau_v}$ is the vegetation-induced shear stress (i.e. drag force per unit surface area) (kg/ms$^2$) and $\rho$ equal to 1000 kg/m$^3$ is the water density.

The bed shear stress is computed using the Manning formulation:

$$\boldsymbol{\tau_b} = \frac{\rho g n^2}{h^{\frac{1}{3}}} \boldsymbol{v}||\boldsymbol{v}|| \qquad (3)$$

where $n$ is the Manning coefficient, which is a calibration parameter accounting for bed roughness (section 3.7). The vegetation-induced shear stress is modelled as the drag force per unit surface area induced by a random or staggered array of rigid vertical cylinders with uniform properties (Baptist et al., 2007; Horstman et al., 2021):

$$\boldsymbol{\tau_v} = \frac{1}{2}\rho C_D a h \, \boldsymbol{v}||\boldsymbol{v}|| \qquad (4)$$

where $C_D$ is the dimensionless bulk drag coefficient and a is the representative density of frontal area per unit depth (m$^{-1}$), which is calculated as:

$$a = DM \qquad (5)$$

where $D$ is the average diameter of prop roots (m) and $M$ is the density of prop roots (i.e. the number of prop roots per unit surface area) (m$^{-2}$).

Both $a$ and $C_D$ refer to the characteristics of the mangrove prop roots. We therefore introduce the mangrove-induced drag coefficient $C_M$:

$$C_M = a C_D \qquad (6)$$

Using $C_D$ equal to 1, as is generally assumed (Baptist et al., 2007), we obtain a value of $C_M$ equal to 3 m$^{-1}$, which is considered here representative for *Rhizophora* mangrove trees (Mazda et al., 1997, 2005).

## 2.3 Model domain and computational grid

The model domain (Fig. 1A) stretches from the continental shelf at the open ocean, corresponding to the seaward limit of the Gulf of Guayaquil, to 50 km upstream along the Daule and Babahoyo river from Guayaquil. Mangrove areas and intertidal mudflats were delineated using remote sensing images (Supplementary section 1). Together these form the intertidal zone and are included in the model domain. To determine the mesh resolution, we followed three approaches depending on the location within the model domain:

   a) At the open sea, cell size ranges from 70 m to 250 m and varies as a function of the bathymetric gradient in order to accurately capture sea bottom topography.
   b) Inside the channels dissecting the delta, we defined the mesh resolution as a function of the channel width, guaranteeing a minimum of 4 nodes per channel cross-section. The resulting mesh resolution ranges between 3 m and 100 m (Fig. 2).



    c)   Inside the mangrove forests, we defined the mesh resolution based on the distance to the channel edge to ensure

150     a smooth transition in mesh resolution from the channel into the mangroves. Resulting cell size ranges from 10

    m near the narrowest channels up to 100 m in the forest interiors (Fig. 2).

The entire mesh consists of 3 212 408 nodes and 6 425 420 elements.

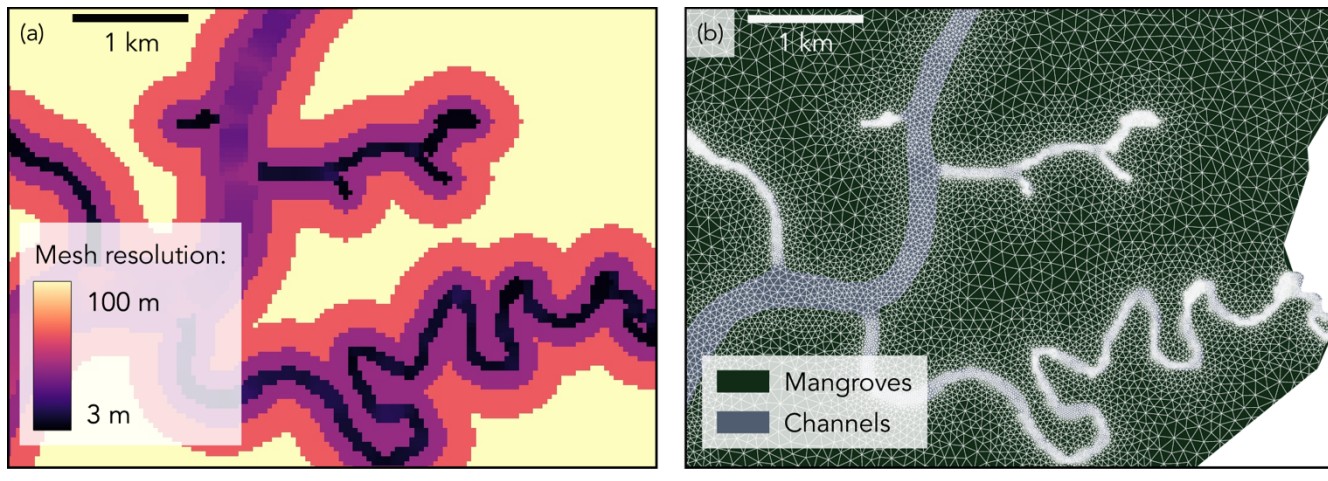

**Figure 2. Illustrative zoom-in of the model domain showing the mesh resolution (a), which in the channels is defined as function of**
155 **the channel width and in the mangroves as function of the distance to the channel edge. The resulting mesh (b) has a mesh resolution**
**as fine as 3 m in the narrowest channels.**

## 2.4 Bathymetry

We obtained bathymetry data of the open ocean from the General Bathymetric Chart of the Ocean (GEBCO) and, inside the

delta, from nautical charts shared by the Oceanographic Institute of the Navy in Ecuador (INOCAR). To estimate the

160 bathymetry on each mesh node, we subdivided the domain into five zones for which we applied a different procedure (Fig. 3):

3 zones in the channels (referred to as level-I, level-II and level-III channels), one in the mangroves and one in the intertidal

flats.



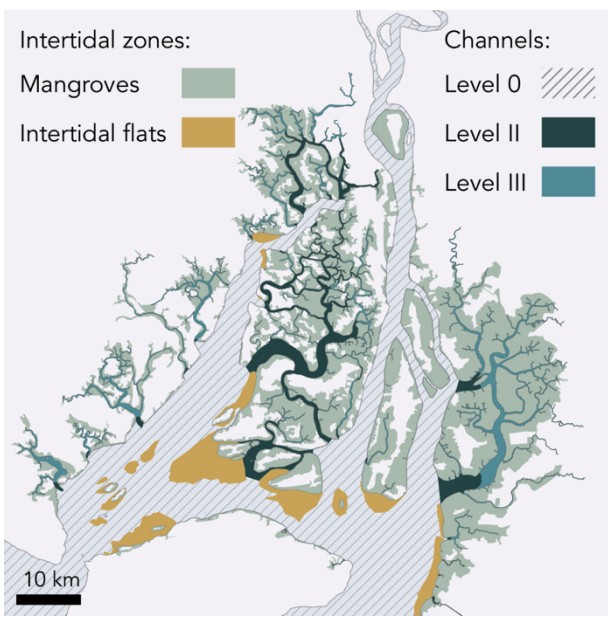

**Figure 3. Overview of zones in the Guayas delta for which we applied a different procedure to define the bathymetry: for the grey dashed area (level-I subtidal channels), dense bathymetric data are available; for the dark blue area (level-II channels, 87 km$^2$), scarcely spread bathymetric data are available; for the light blue area (level-III channels, 348 km$^2$), no bathymetric data are available. Also, no bathymetric data are available for the light-brown area (intertidal flats, 131.8 km$^2$) and the green shaded area (mangrove forests, 974.2 km$^2$).**

In the level-I channels, defined as wide channels, in which the distance between bathymetric observations is much smaller than the channel width, we calculated the bathymetry using a linear triangular irregular network (TIN) interpolation. In the level-II channels, defined as intermediate channels, in which the channel width is smaller than the distance between bathymetric observations, a linear interpolation would lead to disconnected channels with an unrealistic bathymetry (Supplementary section 2). Therefore, for each bathymetric observation point we calculated so-called channel coordinates (Fig. 4A) as the distance to the channel centerline (so-called *perpendicular-centerline coordinate*) and the distance along the channel centerline (so-called *along-centerline coordinate*). The bathymetry on each mesh node was then calculated by a TIN linear interpolation of the bathymetric observations using this channel coordinate system instead of cartesian coordinates (Fig. 4B). In the level-III channels, defined as small channels, in which no data were available, we use a bed elevation of 2 m below the mean low water level during spring tides, in order to guarantee that the channels are always subtidal.





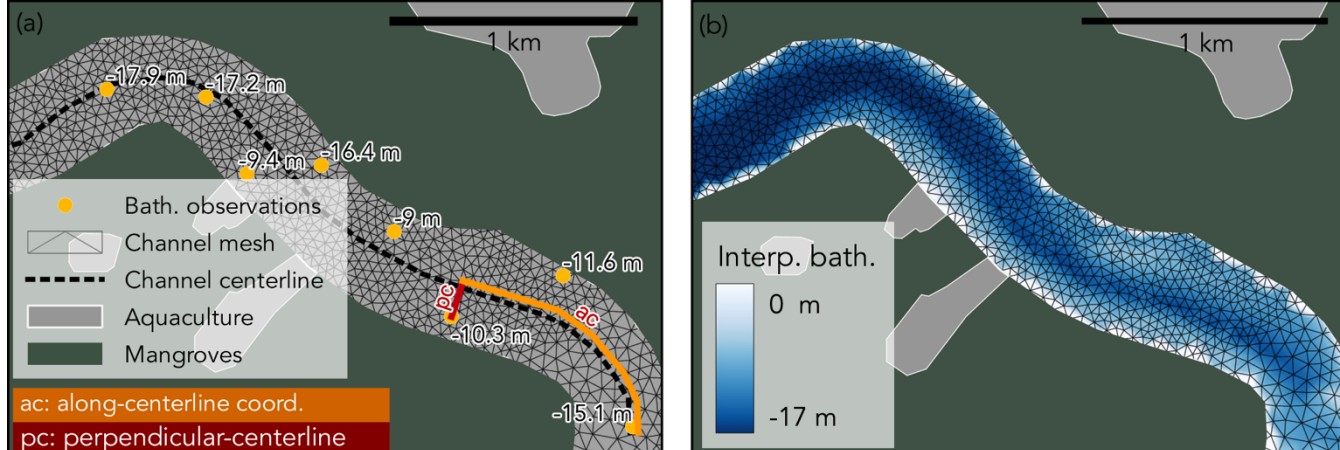

**Figure 4. Example of channel segment where the distance between bathymetric observations is on average larger than the channel width and indicative example of the along-centerline and perpendicular-centerline coordinates (a) and the resulting interpolated bathymetry, which conserved the thalweg and bed shape (b).**

The available bathymetric data do not cover the intertidal flats and mangroves. At all intertidal flats (Fig. 5A), we extracted the waterline (border between submerged and emerged land, Fig. 5B) using the Modified Normalized Difference Water Index (MNDWI) computed from satellite images (ESA Sentinel-2) at different times. For each detected waterline, its water level was then estimated from a water level observation at a nearby tide gauge station. As such, each waterline was considered as an elevation contour line. Based on all contour lines, we interpolated the intertidal flat topography (Fig. 5C, see Supplementary section 3 for more details on the waterline method). The mangrove bed topography was obtained by the model itself after calibrating the Manning coefficient and is therefore described below (Section 2.8).



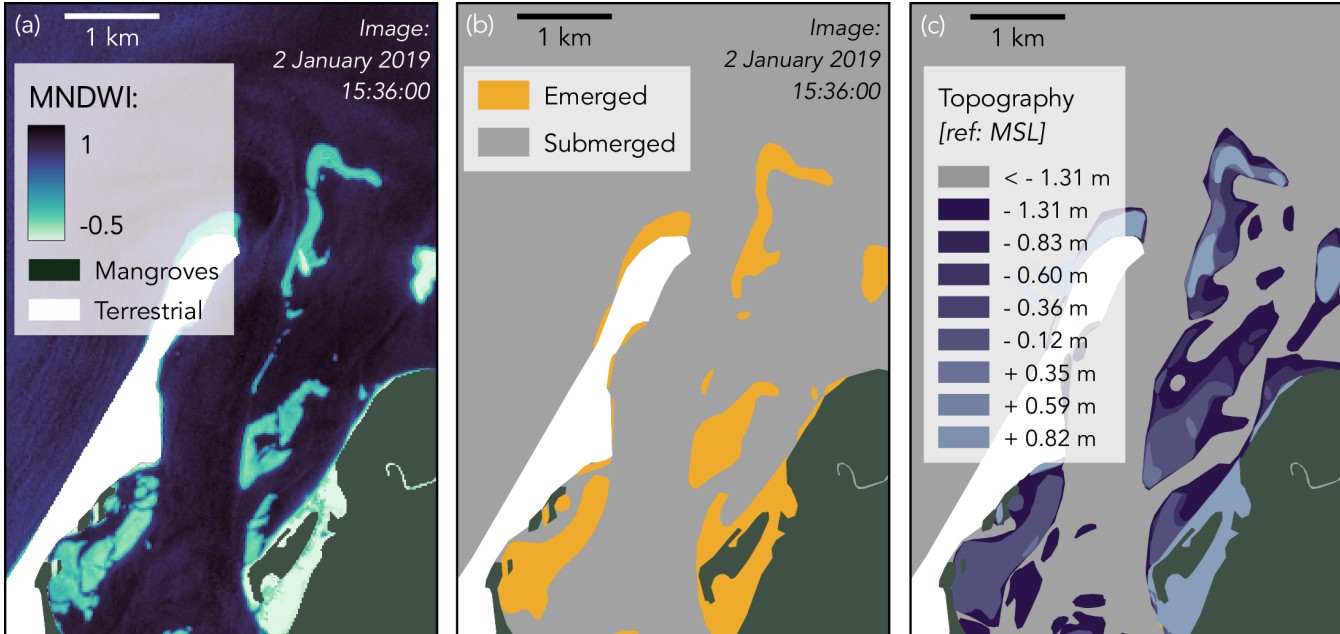

**Figure 5. Zoom in of the Modified Normalised Difference Water Index (a) of a Sentinel 2 image taken on 2 January 2019 15:36:00, based on which we classified a part of the intertidal flat as to be emerged and another part to be submerged (b). The elevation of the waterline (i.e. line between submerged and emerged parts of the intertidal flat) is then estimated from nearby tide gauge data. Together with other images taken at different times, this resulted in a set of elevation contour lines to describe the topography of the intertidal flat (c).**

## 2.5 Boundary Conditions

We derived tidal water levels and velocities at the seaward boundary from the global tidal models TPXO9 (Egbert & Erofeeva, 2002). Upstream river discharge data were obtained through INHAMI (Ecuador's national meteorological and hydrological institute). The available discharge data only represent 73% of the watershed of the Guayas delta but was completed using a linear precipitation-weighted interpolation with monthly precipitation data collected from OpenLandMap (Hengl & Parente, 2022), which is further explained in Supplementary section 4.

## 2.6 Vertical reference level

Local mean sea level (MSL) typically does not coincide with an equipotential surface (i.e., the water surface in rest) along a river delta, but often changes with respect to such a surface due to mechanisms including downstream river discharge and the asymmetry between flood and ebb currents. This sea-to-land gradient in local MSL implies that local MSL can not be used as the vertical reference level for bathymetric and tide gauge data across the whole model domain. Apart from that, the vertical reference surface of the hydrodynamic model is by definition an equipotential surface as gravity only acts in the vertical direction (Slobbe et al., 2013). All collected data (bathymetry and tide gauge data) were originally obtained relative with respect to the local mean sea level (MSL) but referenced to a so-called quasi-geoid model computed from the XGM2019 geopotential model (Zingerle et al., 2020). For further details we refer to Supplementary section 5. To account for any bias



between the observation- and model-derived time series introduced by errors in the vertical referencing, all water level time series were centralised by subtracting the mean.

## 2.7 Model performance metrics

In the model calibration and validation procedure, performance was assessed by comparing water levels in 11 tide gauge stations (Fig. 1), out of which 10 stations have recorded water levels at an interval of one minute and one station at an interval of one hour. In order to quantify the model performance, we calculated the relative tidal range error ($RE$), Nash and Sutcliffe model efficiency ($ME$) (Nash & Sutcliffe, 1970) and the centralised root mean squared error ($CRMSE$):

$$RE = \left|\frac{R_o - R_m}{R_o}\right| \quad\quad (7)$$

$$ME = 1 - \frac{\sum_{i=1}^{N}(o_i - m_i)^2}{\sum_{i=1}^{N}(o_i - \bar{o})^2} \quad\quad (8)$$

$$CRMSE = \sqrt{\frac{\sum_{i=1}^{N}((o_i - \bar{o}) - (m_i - \bar{m}))^2}{N}} \quad\quad (9)$$

where $R_o$ is the observed tidal range (m), defined as the maximum difference between a consecutive low and high water event , $R_m$ is the modelled tidal range (m), $N$ is the total number of observations at a tide gauge station, $o_i$ are the observed water

levels (m), $m_i$ are the modelled water levels (m), $\bar{o}$ is the mean of the observed water levels (m) and $\bar{m}$ is the mean of the simulated water levels (m). Values of $ME$ larger than 0.65 are considered as excellent (Allen et al., 2007). All metrics were calculated on the observed and simulated water level series without centralising.

## 2.8 Calibration of bottom friction

We calibrated the Manning coefficient $n$ to fit observed tidal water levels. To isolate the effects of the Manning coefficient in
the subtidal channels from uncertainties in the intertidal mangrove topography, we first calibrated the Manning coefficient during five high and five low waters around a neap tide (22-24 September 2019), as field measurements showed that mangroves in the Guayas delta do not flood during neap tides (Belliard et al., 2021). We tested Manning coefficient values ranging between 0.0075 and 0.02. We obtained the best model performance with a combination of $n$ equal to 0.0175 in the outer delta and western branch, and $n$ equal to 0.0125 in the eastern branch. More details on the calibration process are given in Supplementary
section 6.

## 2.9 Mangrove platform elevation

Due to lack of data, the mangrove platform elevation ($z_m$) was estimated through calibration. During a spring tide on 29 September 2019, the water depth inside a mangrove forest reached a maximum of approximately 60 cm at three spatially dispersed surveyed locations in the delta, within less than 100 m from a channel edge (Fig. 1) (Belliard et al., 2021).



The mangrove platform elevation was therefore calibrated by iteratively simulating the spring tide on 29 September 2019, while targeting a water depth of 60 cm on every mangrove mesh node adjacent to a channel mesh node. Each simulation resulted in a maximum water level inside the mangrove forests, from which 60 cm was extracted to obtain $z_m$ for a new simulation. Because the input $z_m$ for the new simulation is different, the simulation produced a slightly different maximum water level, which on its turn was used to calculate an updated $z_m$. After 7 iterations, the RMSE of $z_m$ between the two latest

iterations was smaller than 5 cm. The eventual mangrove platform elevation is $z_m$ of the latest iteration (Fig. 6).

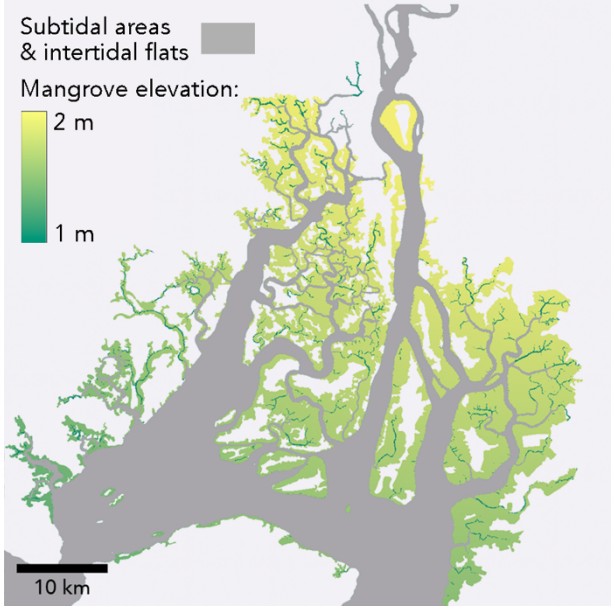

**Figure 6. Mangrove platform elevation ($z_m$) referenced to XGM 2019 for the entire Guayas delta. Elevation values were obtained through a stepwise calibration process in order for simulated water depths to match with observed water depths inside the mangrove forest.**

**2.10 Sensitivity scenarios**

To determine effects of how intertidal area properties are accounted for in the simulation of the delta-scale tidal propagation, we tested the model sensitivity to (1) the inclusion or exclusion of mangroves in the model domain, (2) the mangrove-induced drag, (3) the mangrove platform elevation, (4) the intertidal flats surface elevation, and (5) the explicit bathymetric representation or omission of channels inside the mangroves. For each class of scenarios, we varied the input variable (see

Table 1), and we compared the resulting tidal ranges with the reference simulation.

We analysed scenarios where the mangrove platform elevation $z_m$ was set to 1 m lower, 0.5 m lower, and 0.5 m higher than in the reference simulation. Any higher mangrove bed surface elevation would result in negligible flooding of the mangrove forests, which is a case already covered above (exclusion of mangroves). Any lower mangrove platform elevation would lead to significant flooding of the mangrove forests during neap tides, which is not realistic. To test the impact of exclusion of





mangroves, we have set up a scenario where $z_m$ was set to 10 m above the reference level on the mangrove platforms. As such, the mangroves do not flood, even during a spring tide.

To test the model sensitivity to the presence of channels inside the mangroves, we designed two scenarios: (1) where we replaced level-III channels (Fig. 3) by mangrove platforms and (2) where we replaced both level-II and level-III channels (Fig. 3) by mangrove platforms. The corresponding channel mesh nodes were turned into mangrove mesh nodes by setting the bed

elevation equal to the mangrove platform elevation $z_m$ of the nearest mangrove mesh node, and by applying the mangrove-induced drag value $C_m$ equal to 3 m$^{-1}$.

The mangrove-induced drag in the model is quantified by $C_m$, see eq. (5) and (6), and was set equal to 3 m$^{-1}$ for the reference run (Section 3.1). Here, we simulated tidal propagation for $C_m$ equal to 0 m$^{-1}$ (intertidal wetlands without vegetation), $C_m$ equal to 1 m$^{-1}$ and an extreme value of $C_m$ equal to 25 m$^{-1}$.

Our study area contains large intertidal flat areas (Fig. 3). To test the model sensitivity to the intertidal flat topography $z_f$, we ran scenarios where the intertidal flats were considered to be completely flat. Three elevations for the intertidal flats were tested: -3 m, -1 m and +1 m with reference to XGM 2019.

**Table 1. Overview of the scenarios and which variable is affected in each scenario run.**


| *Classes of scenarios* | *Changed variable* |
| --- | --- |
| *Reference* | $C_m = 3$ |
| | $z_m \rightarrow$ see section 2.9 |
| | $z_f \rightarrow$ see section 2.4 |
| *Mangrove platform elevation* | $z_m = -1$ m |
| | $z_m = -0.5$ m |
| | $z_m = +0.5$ m |
| | $z_m = +10$ m (no mangroves) |
| *Small channels* | Level I $\rightarrow$ closest $z_m$ |
| | Level I + II $\rightarrow$ closest $z_m$ |
| *Mangrove-induced drag* | $C_m = 0$ m$^{-1}$ |
| | $C_m = 1$ m$^{-1}$ |
| | $C_m = 25$ m$^{-1}$ |
| *Tidal flat topography* | $z_f = -3$ m |
| | $z_f = -1$ m |
| | $z_f = +1$ m |



## 3. Results

### 3.1 Model validation

To evaluate the model performance, we compare simulated water levels with observed water levels along the eastern and western branches, over three tidal waves. During a spring tide on 29 October 2019 (Figure 7 and Figure 8), both observations and simulation show tidal amplification with an observed increase in tidal range of 24 cm in the eastern branch (from station 9 to station 7) and 126 cm in the western branch (from station 5 to station 1). The *RE* ranges from 1.2 % (station 8) to 6.3 % (station 1), *RMSE* is 0.18 ± 0.09 m (average +/- standard deviation for 10 stations, water level series from one station did not

cover the validation time span) and *ME* is 0.85 ± 0.10. The latter average value indicates excellent performance (Allen et al., 2007). In the eastern branch, there is a good agreement between simulated and observed tidal range. During a neap tide on 6 November 2019, tidal range increased with 53 cm and 65 cm in the western and eastern branches respectively. *RE* ranges from 4.7 % (station 3) to 9.1 % (station 7), *RMSE* is 0.11 ± 0.03 m (average +/- standard deviation for all 11 stations) and *ME* is 0.60 ± 0.32.

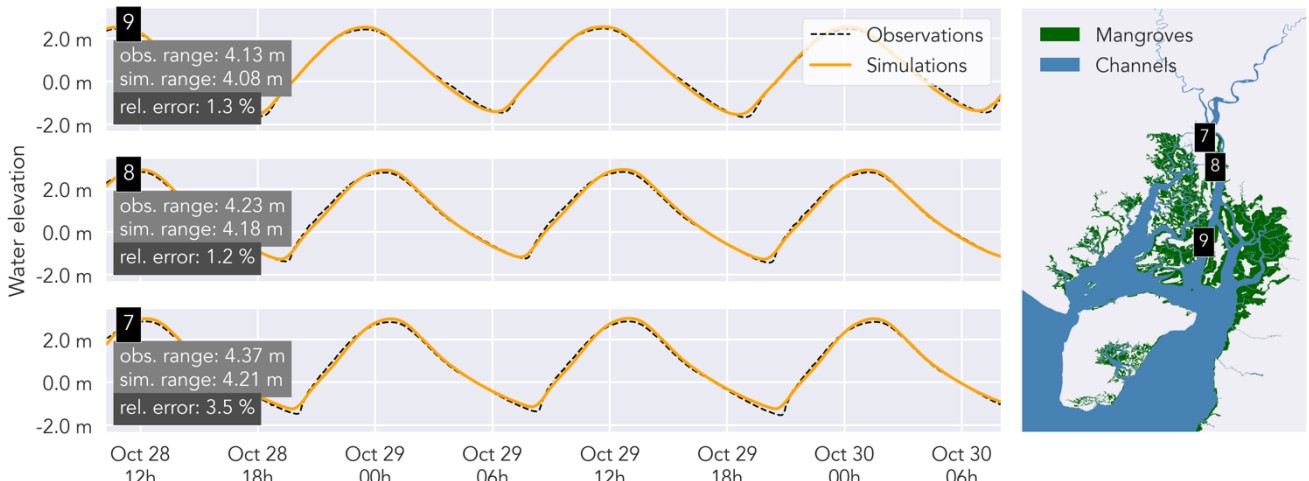


**Figure 7. Observed (dashed black line) vs. simulated water levels (orange lines), plotted for three stations in the eastern branch. The relative error is calculated as the observed tidal range minus the simulated range, divided by the observed tidal range. Observed and simulated water levels are centralized (mean water elevation was subtracted) compared on a 1-minute timestep.**




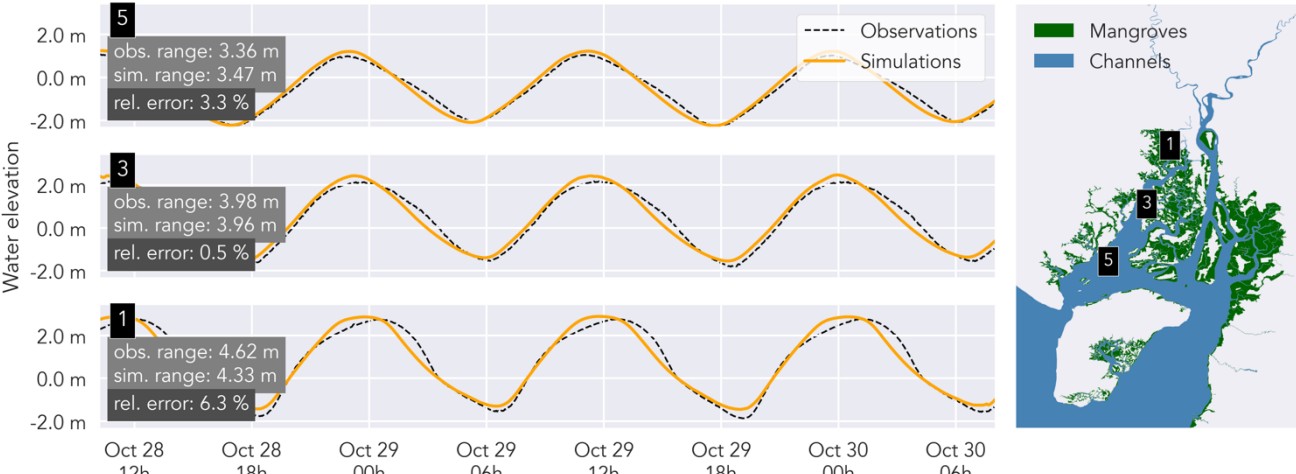

**Figure 8. Observed (dashed black line) vs. simulated water levels (orange lines), plotted for three stations in the western branch. The relative error (rel. error) is calculated as the observed tidal range minus the simulated range, divided by the observed tidal range. Observed and simulated water levels are centralized (mean water elevation was subtracted) compared on a 1-minute timestep.**

### 3.2 Model sensitivity

Among the tested scenarios, the scenarios with varying mangrove elevation result in the largest variety between simulated high water levels (Figure 9A and Figure 10A). The scenarios with mangrove platform elevation + 50 cm and no mangroves (Table 1) result in high water levels up to 22 cm and 29 cm higher than the reference scenario, respectively, upstream in the western branch. For the scenarios with mangrove elevation -50 cm and -100 cm high water levels are respectively 21 cm and 39 cm lower than the reference scenario, upstream in the western branch.

The scenarios with varying degree of channelization result in the second largest variation in simulated high water levels (Figure 9B and Figure 10B). For the scenarios with different degrees of channelization (Table 1), upstream high water levels in the western branch differ up to 22 cm and 12 cm when level-II and III channels and only level-III channels are omitted, respectively.

Compared to the mangrove platform elevation and the degree of channelization, the mangrove-induced drag has a smaller impact on the distribution of high water levels, especially in the western branch (Figure 9C and Figure 10C). At the most upstream considered point of the eastern branch, the scenarios with mangrove-induced drag coefficients of 0 and 25 m$^{-1}$ (Table 1) result in respectively 16 cm lower and 5 cm higher high water levels than the reference high water levels.

The differences in high water levels among intertidal flat topography scenarios are smaller than for any other set of scenarios (Figure 9D and Figure 10D). Varying the intertidal flat topography from -3 m to +1 m (referenced to XGM 2019) results in 7 cm high water levels difference in the western branch and 11 cm in the eastern branch.



**Figure 9. High water levels, vertically referenced to XGM 2019, along a longitudinal transect in the eastern branch (e) during a spring tide: scenarios with varying mangrove platform elevation $z_m$(a), varying degrees of channelization (b), varying mangrove-induced drag coefficient $C_M$(c) and varying intertidal flat topography $z_f$(d).**





**Figure 10. High water levels, vertically referenced to XGM 2019, along a longitudinal transect in the western branch (e) during a spring tide: scenarios with varying mangrove platform elevation $z_m$ (a), varying degrees of channelization (b), varying mangrove-induced drag coefficient $C_M$(c) and varying intertidal flat topography $z_f$(d).**

Much larger water volumes are flowing to and from the mangroves for the scenarios with lower platform elevation compared to scenarios with lower mangrove-induced drag. Flow rates become positive earlier in case of lower mangrove platform elevation, indicating an earlier flooding of the mangroves (Figure 11A). Also, the draining of the mangroves back into the channels lasts longer. Not only the total duration during which the mangroves flood and drain increase with lower mangrove platforms but also peak flow rates increase. For the scenarios with varying mangrove-induced drag, the peak flow rate also differs significantly with the highest flow rates reached for the lowest $C_m$ values (Figure 11B). However, the start and the end of the wetland flooding is the same for all scenarios with varying $C_m$ values, in contrast to scenarios with varying mangrove platform elevation.





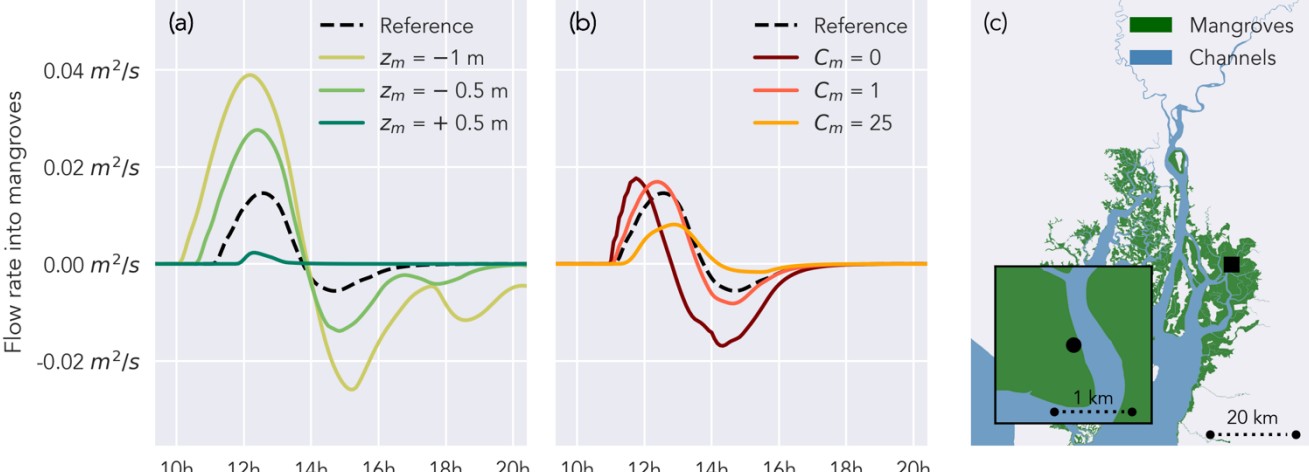

**Figure 11. Flow rates from the channel into the mangroves, expressed as discharge per meter of channel-mangrove boundary for a simulated spring tide on 30 September 2019. Scenarios with varying mangrove platform elevation $z_m$ (a) and with varying mangrove-induced drag $C_M$ (b). Flowrates are shown for a representative location in the delta (c). Positive values indicate flow from the channel into the mangroves and vice versa. The area below each line gives an indication of the total volume of water flowing to and from the mangroves during flood and ebb tides, respectively.**

## 4. Discussion

Current knowledge on how mangroves can attenuate high water levels in large-scale deltas was restricted to either (1) model cases which did not explicitly capture the complex geometry of channels intertwined by intertidal wetlands (Chen et al., 2021; Deb & Ferreira, 2017; Zhang et al., 2012) or (2) small-scale observation and modelling studies which only quantified attenuation inside mangroves (Horstman et al., 2021; Montgomery et al., 2018). Modelling studies in tropical areas are typically restricted due to the scarce availability of data on channel bathymetry and mangrove topography. Therefore, we still have a limited knowledge on high water levels propagation through a large delta and especially on the role of mangrove vegetation properties and mangrove topography.

Here, we have presented a delta-scale model of the Guayas delta which captures the propagation of high water levels during a spring tide, despite limited data availability. Calibration and validation are based on water levels, similar to previously published large-scale coastal models (Chen et al., 2021; Stark et al., 2016; Zhang et al., 2012). Through a series of scenario analyses, we show that: (1) mangrove elevation and the presence or absence of mangroves is more important than mangrove-induced drag in determining high water levels across the delta; (2) increasing or decreasing the elevation of intertidal flats, located near the downstream end of the delta, has little effect on upstream high water levels; and (3) the degree of channelization inside the mangrove forests determines high water levels both upstream and downstream. These findings are further discussed in detail below.





### 4.1 Effect of mangrove-induced drag and mangrove platform elevation

Our results reveal that upstream HWLs increase with increasing mangrove drag coefficient and vice-versa (Figure 9C and Figure 10C). A lower drag in the mangroves allows a larger fraction of the tidal prism to flow from the channels into the mangroves during flood tides. While with higher mangrove-induced drag, the fraction of water which is conveyed through the channel increases (Figure 11). However, largely different values of the mangrove-induced drag are shown to result in relatively small differences in high water levels (Figure 9C and Figure 10C). This low sensitivity of along-channel high water level

attenuation to differences in the vegetation-induced drag in fringing mangrove forests, is confirmed on the small scale (~0.1 km$^2$) by Horstman et al. (2015), where they attribute this to the low flow velocities inside mangroves and consequently a low drag term in in the shallow water equations. Also for tidal marshes, Hu et al. (2015) confirmed that variations in stem density have little effect on variations in water currents in vegetated wetlands. However, only sub-canopy drag is considered here. Chen et al. (2021) confirms the relatively small role of mangrove density on high water levels reduction; however, they argue

that if high water levels reach the top of the mangrove canopy, the drag strongly increases and thus, vegetation properties such as tree height could still play a role if water levels would exceed the canopy height. Our results suggest that on the delta-scale, high water levels are much more sensitive to mangrove platform elevation than mangrove vegetation properties (Figure 9A and Figure 10A). This is due to the higher sensitivity of flow towards the mangroves for mangrove platform elevation compared to mangrove-induced drag (Figure 11).

### 4.2 Effect of intertidal topography

Simulated high water levels appeared to be much more sensitive to changes in mangrove platform elevation than changes in intertidal flat elevation, although the tested range of intertidal flat elevation, 4 m, was much larger than the range of mangrove platform elevation, 1.5 m (Figure 9D and Figure 10D). The total intertidal flat area in the delta is much (7.4 times) smaller than the total area of mangroves. Therefore, lowering the intertidal flats with 3 m will still result in less extra flood storage

volume than lowering the mangroves with, for instance, 0.5 m. However, Li et al. (2012) ran similar scenarios with or without tidal flats in Darwin Harbour (Australia), where intertidal flat area is much more similar to mangrove area, and still found a greater effect on excluding the mangroves compared to excluding the tidal flats. An additional factor in our case is that most of the intertidal mudflat area is located near the downstream end of the delta, while mangroves also exist much more landward (Figure 3). Smolders et al. (2015) has demonstrated that upstream-located wetlands have a larger effect on along-channel

attenuation because estuarine channels typically narrow in the upstream direction and consequently, wetlands of the same surface area but located more upstream, can accommodate a larger portion of the landward propagating flood water volume.

### 4.3 Effect of degree of channelization inside the mangrove forest

Scenarios where we partly removed channels inside the mangrove forests, led to higher high water levels both upstream and downstream from the considered channels (Figure 9B and Figure 10B). These channels are mostly side branches of the main,



large estuarine channels, which run from the main channels into the mangrove forests, while further branching, narrowing and ultimately ending in the forests (Figure 3). Hence these channels act as rapid conduits for flood high water levels propagation from the main estuarine channels into the mangrove forests, and flood propagation through such channels is more rapid as compared to vegetated mangrove platforms (Horstman et al. 2015) or marsh platforms (Vandenbruwaene et al. 2015). Therefore, the presence of channels inside mangroves is typically considered to lower the within-wetland attenuation capacity

of a mangrove forest (Horstman et al., 2021; Krauss et al., 2009; Montgomery et al., 2018). However, here we show that on the delta-scale, the presence of branching channel networks inside the forests leads to lower upstream high water levels and hence higher along-channel attenuation. This can be explained as the channels running into the mangroves act as an efficient conveyance of water out of the main channel into the mangroves and therefore, allow a larger fraction of the flood water volume to spread out into and to be stored temporarily in the mangroves. Consequently, high water levels in the main channels

are lowered.

### 4.4 Implications for modelling high water levels in data-scarce deltas

Recently, modelling studies on high water level propagation have primarily focused on the detailed representation of drag in relation to the mangrove vegetation structure (Chen et al., 2021; Montgomery et al., 2019; Yoshikai et al., 2021). However, we demonstrate here that it was more important to include detailed representation of the channel networks inside mangroves,

mangrove platform elevation and mangrove spatial extent. In general, mapping topography in mangrove forests is challenging due to the dense canopy cover prohibiting the use of highly accurate GPS surveying or remote sensing (Gijsman et al., 2021). Here we demonstrated a method to estimate mangrove platform elevation, based on measurements of water depth inside the mangroves and calibrating the mangrove platform elevation so that observed water depths are reproduced by the model. This procedure enabled us to fill the gap in data availability and to model high water levels propagation on a delta scale. However,

the spatial coverage of our water depth observations is limited to 3 locations and only close to the channels. Obtaining a denser network of water depth measurements which are spread more equally over the entire delta, and which capture water depths deeper in the forest would further improve the calibration of the mangrove platform elevation. While remote sensing is insufficient to map below-canopy topography, it is still valuable to map channel networks as the latter is proven here to play an important role in conveying water from the channels into the mangrove forest. By detecting creeks and channels from

satellite pictures, however, small channels which are covered by the overhanging canopy are not included. Nevertheless, the presence of such small channels in mangroves is limited (Schwarz et al., 2022), which may suggest that the role of such small channels for delta scale flood propagation is likely to be less significant. Furthermore, while we have found that the topographic representation of intertidal mudflats barely affected the simulation of high water levels in our study case, this might play a more important role in delta systems where intertidal flats cover larger portions of the delta and/or are located more upstream.

In such cases, the applied waterline technique is suitable to derive the topography of intertidal flats (Bishop-Taylor et al., 2019; Zhang et al., 2022), as it is based on freely available satellite imagery and requires limited workload.

### 4.5 Implications for nature-based flood risk mitigation

Our model results stress that for mangroves to serve as effective nature-based risk mitigation in river deltas, vast areas of mangrove forest are necessary rather than densely vegetated forests. With the recently increasing recognition of mangroves as

a natural flood buffer (Temmerman et al., 2022), mangrove restoration projects have become more and more popular (Su et al., 2021). Our results imply that young, restored mangroves, together with naturally expanded young mangroves, could immediately contribute to upstream high water levels reduction, even before developing into a fully matured mangrove forest. In addition, the presence of an extensive channel network inside mangroves would also increase the effectiveness of a mangrove forest to temporarily reduce peak water levels during extreme sea level events, and hence to lower flood risks on

the large delta scale.

### Author contributions

IP, JPB, LD, ST and OG contributed to the design of the study and collecting the necessary data and CS contributed in implementing the vertical reference level. IP and OG performed the model setup with contributions and feedback from JPB and ST. IP wrote the first draft of the manuscript with contribution of CS on the description of the geoid. All authors contributed

to writing and revising the manuscript and approved the submitted version.

### Competing interests of Interest

The authors declare that they have no conflict of interest.

### Acknowledgments

We thank Jan De Nul and INOCAR for sharing their tide gauge data and INHAMI for sharing river discharge data. The

computational resources and services used in this work were provided by the HPC core facility CalcUA of the Universiteit Antwerpen, and VSC (Flemish Supercomputer Center), funded by the Research Foundation - Flanders (FWO) and the Flemish Government. Furthermore, we would like to thank the Research Foundation Flanders (FWO, Belgium) for the PhD fellowship for fundamental research for I. Pelckmans (11E0723N). J.-P. Belliard is supported by FED-tWIN ABioGrad. O. Gourge was supported by the European Union's Horizon 2020 research and innovation program under the Marie Skłodowska-Curie grant

agreement No 798222. The study was locally supported in the context of the VLIR-UOS Ecuador Biodiversity Network project.



**Data Availability**

All model results (water levels) are available upon request with the authors.

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
