# Peer review of "Mangrove ecosystem properties regulate high water levels in a river delta"

_EGUsphere, 2023_

## Author Response (AR1)

Reply letter to EGUSPHERE-2023-428 | NHESS
**Mangrove ecosystem properties regulate high water levels in a river delta**

Dear Associate Editor,
Dear Dhruvesh Patel,

Please find below a point-by-point response to all referee comments. Revised paragraphs of the manuscript are reproduced with tracked changes (new text, removed text) which can be found in the revised manuscript and supplementary materials.

We hope you appreciate our response to their constructive remarks, and that you will consider this revised version for publication in Natural Hazards and Earth System Sciences.

We would like to thank you and the referees for the suggestions and improvements of our manuscript.

Kind regards,
Ignace Pelckmans

Response to referee 1.

**Comment 1:**
*Sentinel-2 is known for providing fine spatial resolution multispectral images. However, the paper fails to address the spatial resolution discrepancy between the green and SWIR bands of Sentinel-2, which are 10 m and 20 m, respectively. This raises an important issue when calculating the Modified Normalized Difference Water Index (MNDWI) from Sentinel-2 images. Two options are available: (1) Upscaling the 10-m green band to 20 m and calculating MNDWI at 20-m resolution, which is simpler but sacrifices detailed information from the 10-m resolution. (2) Downscaling the 20-m SWIR band to 10 m using a pan-sharpening technique and calculating MNDWI at 10-m resolution, which is more complex but preserves the 10-m information and yields a finer result. However, neither of these methods is discussed or mentioned in the paper. It is essential to address this issue and provide a rationale for the chosen approach in calculating MNDWI from Sentinel-2 images.*

This is indeed not described in the manuscript or supplementary materials, but we agree that it is an important step in the procedure which should be described. In the methods section of the main manuscript our goal is to give an overview of the various techniques we used to overcome the data scarcity challenge. For the technical explanation of these methods, we refer to the supplementary materials. Therefore, we suggest adding to Supplementary section 3:
The spatial resolution of the Green and SWIR bands of Sentinel 2 imagery have a spatial resolution of 10 and 20 m respectively. When calculating MNDWI, the resolution of the Green band was scaled to 20 m using an average-based aggregator. Consequently, the eventual spatial resolution of MNDWI was 20 m. As > 99 % of the mesh elements within the intertidal flats have a resolution coarser than 20 m, generating the intertidal topography at a spatial resolution of 20 m was considered sufficient.

Response to referee 2.

**General remark:**
"Overall the quality of the manuscript is acceptable, and I would accept it with some minor revisions."

**Comment 1:**
Please justify the choice: "We defined the mesh resolution as a function of the channel width" (L146). Citation required.

We defined the narrower the channel, the finer the mesh resolution. This was to guarantee a minimum of 4 nodes per channel cross-section to ensure good connectivity within the channels. This means that for a 20 m wide channel, mesh resolution is as small as 5 m. For much wider channels (100s of meters) there is no need for such a fine resolution which would make the model computationally heavier. Hence, by expressing the mesh resolution in function of channel width we can guarantee channel connectivity without dramatically increasing the computation requirements. A similar approach is presented by Deb et al. 2022.

In the main manuscript, we propose to adapt the statement on mesh resolution in function of mesh resolution (L146): Inside the channels dissecting the delta, we defined the mesh resolution as a function of the channel width,

with a minimum of 4 nodes per channel cross-section to guarantee channel connectivity (Deb et al. 2022). The resulting mesh resolution ranges between 3 m and 100 m (Fig. 2).

**Comment 2:**
Cite the similar practice for " L149 we defined the mesh resolution based on the distance to the channel edge to ensure a smooth transition in mesh resolution from the channel into the mangroves."

Similar to comment 1, we enforced this rule to guarantee a smooth transition between the channels with fine mesh resolution to the mangrove platform where mesh resolution is coarser. A similar approach is presented by Deb et al. 2022.

In the main manuscript, we propose to add that we followed a similar practice as in Deb et al. 2022 (L149).

**Comment 3:**
L289 NSE (ME) 0.60 ± 0.32 is given; what was typically accepted range in the similar study or model perforomnce accepted for decision making.

We acknowledge that we should refer to commonly accepted ranges of NSE model efficiency. In section 2.7, we already describe that ME > 0.65 are considered excellent (Allen et al. 2007) but we will extend this by referring to (Moriasi et al. 2007; Gori et al., 2020) who considered ME > 0.5 as acceptable. We propose to also refer to these references at the end of section 3.1 (L289).

**Comment 4:**
How this study can be implemented to other data scarce costal regions of the world, a little discussion will be worthwhile.

I suggest to add this paragraph to section 4.4:

28 million people living in developing or least developed economies are prone to coastal flooding due to tropical storms only (Edmonds et al. 2020). In addition, long-term climatic fluctuations such as ENSO can also lead to extreme sea level events and related coastal hazards, especially in developing countries in the tropics (Belliard et al. 2021; Reguero et al. 2015). Nevertheless, the majority of efforts in modelling high water levels in deltas and estuaries are concentrated in temperate regions (e.g. Stark et al. 2016; Smolders et al. 2015; Lawler et al. 2016, Sheng et al. 2021; Harrison et al. 2021) or in developed tropical countries (e.g. Zhang et al. 2012, Li et al. 2012, Liu et al. 2013; Dominicis et al. 2023). Here, we demonstrate how freely available data can contribute to filling this gap in geographical coverage of deltaic high water level modelling. Delineating the channel- and mangrove extent and mapping unvegetated intertidal topography is based on freely- and globally available Sentinel 2 data, and scarcely spread bathymetric observations can be partly compensated by interpolating along channel coordinates. Nevertheless, there is still a need for vertically referenced water level observations (such as by tide gauges) to apply the waterline method and calibrate and validate simulated water levels. However, future developments in remote sensing such as the recent launch of the SWOT mission by NASA (Biancamaria et al. 2016), might contribute by globally mapping water surface elevations (and water surface slopes). Future research, supported by the presented methodology in this paper together with the current and future availability of free global remote sensing data, should cover a wide variety in river deltas to further develop the potential of conserving wetlands as a nature-based solution for coastal flooding in river deltas.

References

Biancamaria, S., Lettenmaier D. P., and Pavelsky, T. M.: The SWOT mission and its capabilities for land hydrology, 117-147, 2016.

Deb, M., Abdolali, A., Kirby, J. T., and Shi, F. : Hydrodynamic modeling of a complex salt marsh system: Importance of channel shoreline and bathymetric resolution. Coastal Engineering, 173, https://doi.org/10.1016/J.COASTALENG.2022.104094, 2022.

Edmonds, D. A., Caldwell, R. L., Brondizio, E. S., and Siani, S. M. O.: Coastal flooding will disproportionately impact people on river deltas, Nat Commun, 11, 4741, https://doi.org/10.1038/s41467-020-18531-4, 2020.

Gori, A., Lin, N., & Smith, J.: Assessing Compound Flooding From Landfalling Tropical Cyclones on the North Carolina, Coast. Water Resources Research, 56, 4, https://doi.org/10.1029/2019wr026788, 2020.

Harrison, L. M., Coulthard, T. J., Robins, P. E., and Lewis, M. J.: Sensitivity of Estuaries to Compound Flooding, Estuaries Coasts, 45, 1250–1269, https://doi.org/10.1007/s12237-021-00996-1, 2022.

Lawler, S., Haddad, J., and Ferreira, C. M.: Sensitivity considerations and the impact of spatial scaling for storm surge modeling in wetlands of the Mid-Atlantic region, Ocean Coast. Manag., 134, 226–238, https://doi.org/10.1016/j.ocecoaman.2016.10.008, 2016.

Moriasi, D. N., Arnold, J. G., Liew, M. W. V., Bingner, R. L., Harmel, R. D., & Veith, T. L.: Model Evaluation Guidelines for Systematic Quantification of Accuracy in Watershed Simulations, Transactions of the ASABE, 50, 3, 885–900, https://doi.org/10.13031/2013.23153, 2007.

Reguero, B. G., Losada, I. J., Díaz-Simal, P., Méndez, F. J., and Beck, M. W.: Effects of Climate Change on Exposure to Coastal Flooding in Latin America and the Caribbean, Plos One, 10, e0133409, https://doi.org/10.1371/journal.pone.0133409, 2015.

Sheng, Y. P., Rivera-Nieves, A. A., Zou, R., and Paramygin, V. A.: Role of wetlands in reducing structural loss is highly dependent on characteristics of storms and local wetland and structure conditions, Sci. Rep., 11, 5237, https://doi.org/10.1038/s41598-021-84701-z, 2021.